# Effect of Coronavirus 19 on Maintaining Balance in Skilled Athletes

**DOI:** 10.3390/ijerph191710470

**Published:** 2022-08-23

**Authors:** Jarosław Jaszczur-Nowicki, Tomasz Niźnikowski, Hubert Makaruk, Andrzej Mastalerz, Jared Porter, Michał Biegajło, Ewelina Niźnikowska, Logan Markwell, Marta Nogal, Weronika Łuba-Arnista, Paweł Arnista, Oscar Romero-Ramos

**Affiliations:** 1Department of Tourism, Recreation and Ecology, Faculty of Geoengineering, University of Warmia and Mazury in Olsztyn, 10-719 Olsztyn, Poland; 2Department of Gymnastics, Faculty of Physical Education and Health, Józef Piłsudski University of Physical Education, 21-500 Biala Podlaska, Poland; 3Department of Biomedical Sciences, Faculty of Physical Education, Józef Piłsudski University of Physical Education, 00-968 Warsaw, Poland; 4Department of Kinesiology, Recreation, and Sport Studies, The University of Tennessee, Knoxville, TN 37996-2700, USA; 5Department of Tourism and Recreation, Faculty of Health Sciences, John Paul II University of Applied Sciences in Biala Podlaska, 21-500 Biala Podlaska, Poland; 6Department of Physical Education, Faculty of Health Sciences, Lomza State University of Applied Sciences, 18-400 Łomża, Poland; 7Department of Didactics of Languages, Arts and Sports, Institute of Sports, University of Malaga, 4, 29017 Málaga, Spain

**Keywords:** sports, COVID-19, pandemic, athlete, handball players, the Romberg test

## Abstract

Preliminary studies have reported that motor control is negatively impacted following an infection of COVID-19. The purpose of this study was to evaluate the effect of COVID-19 on maintaining balance in highly skilled athletes. As part of a larger investigation that was initiated in 2019, twelve professional handball players were recruited to participate in a study that was designed to measure static balance performance. Following the initial pre-test, six participants (body height 184.8 ± 4.7 cm; body weight 85.5 ± 3.3 kg; age 21.3 ± 1.2 years) were infected with COVID-19. The remaining six participants (body height 188.7 ± 2.6 cm; body weight 92.3 ± 3.7 kg; age 26.3 ± 3.3 years) never tested positive for COVID-19 and were presumably not infected with the virus. The experimental design required all the participants to complete an initial balance assessment (pre-test) and a later balance assessment (post-test). To fully analyze our data, we conducted a 2 (condition: COVID, no-COVID) X 2 (test: pre-test, post-test) ANOVA with repeated measures on the second factor. Our analysis revealed that the skilled athletes who contracted COVID-19 had a significant decrease in balance performance from the pre-test that occurred prior to being infected with COVID-19 relative to the post-test that occurred following the COVID-19 infection. Additionally, the skilled athletes who were not infected with COVID-19 did not demonstrate the same deterioration in balance performance in the same period. This study highlights the impact COVID-19 has on static balance performance in a group of highly skilled handball players. Longitudinal studies are needed to fully understand the lasting impacts COVID-19 has on motor behavior.

## 1. Introduction

COVID-19 has paralyzed and made a permanent mark on the daily lives of many people. To this day, COVID-19 wreaks havoc in many countries around the world and has altered the development of industry and the global economy, in addition to affecting the world of sports. The ways in which major sporting events were held have changed. Specifically, some events have been played within spectator-less arenas, and tests for SARS-CoV-2 (i.e., COVID-19) coronavirus have had to be consistently performed. As further evidence of the way COVID-19 has impacted sports, the Olympic Games were canceled in 2020 and moved to the following year 2021 [1,2]. Following the outbreak of COVID-19 in Wuhan, China, the virus rapidly became a global threat to human health, with significant neurological symptoms being reported [3,4,5]. According to Mulcahey et al. [6], the long-term effects of this virus are currently unclear; therefore, it is important to understand how the potential negative side effects of COVID-19 may affect athletes and what considerations need to be made in return-to-play preparticipation examinations. It is important to note that the understanding of COVID-19 is continually changing as researchers and medical practitioners learn more about this novel virus.

Recent research has shown that athletes had significant problems returning to peak performance following a COVID-19 infection [6,7]. For example, Celik and colleagues [7] found that athletes who were infected by COVID-19 had significantly lower inspiratory and expiratory muscle strength compared to non-infected athletes. Not only has COVID-19 been found to affect physical health, but it is also proposed to negatively impact the long-term mental health of athletes; it potentially contributes to depression and premature termination of a career in competitive sports [7]. Many scientists are trying to determine how COVID-19 affects individual systems of the human body, and what effects it has on further functioning in everyday life as well as in sports. Sensory symptoms, such as loss of taste and smell, have been shown to be common side effects of COVID-19 [4,8,9,10,11,12,13]. Hearing and balance symptoms associated with COVID-19 have also been reported; however, these have not been thoroughly studied [8]. Hearing loss and balance disorders often result from viral infections in the inner ear; however, the effects of COVID-19 on the ear remain poorly understood. Recent research has shown that COVID-19 can enter the inner ear and infect specific cells, which might influence the associated problems with hearing and balance [8]. Specifically, three possible mechanisms have been proposed for these audio-vestibular symptoms observed within COVID-19 patients. This virus may directly infect and kill primary afferent auditory and/or vestibular nerve cells (e.g., relay information to the brain), vestibular hair cells (e.g., detect head movement, acceleration, and deceleration), and cochlear hair cells (e.g., detect sound). Thus, through these potential mechanisms, COVID-19 could lead to impaired hearing and balance.

Balance is defined as the ability to maintain the body’s center of gravity within its base of support and can be divided as either static or dynamic balance. Static balance is the ability to sustain the body in static equilibrium or within its base of support [14]. Dynamic balance is said to be more challenging because it requires the ability to maintain equilibrium during a transition from a dynamic to a static state [15]. Both static and dynamic equilibrium require the integration of visual, vestibular, and proprioceptive inputs to produce an efferent response to control the body within its base of support [16]. That proprioception is the reception of stimuli produced within an organism, whereas balance is physical equilibrium. This implies that proprioception is a neurological process, while balance is the ability to remain in an upright position [17].

Jumping, landing, and pivoting activities involved in team sports have been described as injury risk factors by many authors proposing prevention strategies based on improving athletes’ balance [18,19]. In addition, literature related to rehabilitation states that balance exercise programs may improve proprioceptive functions not only during rehabilitation, but also during a competition period; thus, this protects athletes against injuries effectively [20,21,22]. There are limited data on the influence of balance training on the motor skills of elite athletes. Balance was related to competition levels in some sports, with more proficient athletes displaying greater balance. There were significant relationships between balance and the number of performance measures [23].

Apart from impaired balance potentially due to the observed COVID-19 audiovisual symptoms, a compromised central and/or peripheral nervous system due to COVID-19 could also contribute to such neurological symptoms [4,24]. Neurological symptoms have been reported in individuals with COVID-19, which is not surprising due to the neurological complications observed in previous coronaviruses (e.g., OC-43, 229E, MERS) [25]. The current evidence would suggest that these symptoms are likely due to direct invasion of the neural tissues or indirectly caused by inflammatory reactions within the peripheral nervous system. These impairments have not only been found during the active disease, but have also been seen as post-neurological problems following the infection recovery [26,27]. These impairments can potentially hinder balance and are prevalent given that they have been self-reported in nearly one-third of COVID-19 patients [12]. However, objectively assessing balance before and after a COVID-19 infection has yet to be investigated.

Therefore, there is an immediate need for scientific research to provide objective data on the health of highly trained athletes in various sports disciplines. Given the scarcity of research examining balance following a COVID-19 infection, this study aimed to evaluate the balance of professional athletes who had been diagnosed with COVID-19. Given the previous research, we hypothesized that athletes who had COVID-19 would have impaired balance compared to their balance prior to a COVID-19 infection.

## 2. Materials and Methods

### 2.1. Participants

Following the initial pre-test, six participants (body height 184.8 ± 4.7 cm; body weight 85.5 ± 3.3 kg; age 21.3 ± 1.2 years) were infected with COVID-19 (group E). Six participants were previously diagnosed with COVID-19 by a medical practitioner, which was confirmed by RT-PCR assay. The remaining six participants (body height 188.7 ± 2.6 cm; body weight 92.3 ± 3.7 kg; age 26.3 ± 3.3 years) never tested positive for COVID-19 and were presumably not infected with the virus (group C). The experimental design required all the participants to complete an initial balance assessment (pre-test) and a later balance assessment (post-test). Between the two assessments, players were not allowed to train. Six participants recovered from the infection at the time of their participation in the present study. Specifically, none of the participants had an active infection of COVID-19 while they participated in our study. All the participants were not injured and trained in a high-performance Academic Sports Club; they completed an average of five training sessions and one competitive match per week.

### 2.2. Study Design

This study investigated the effect a COVID-19 infection had on the static balance of the players of a professional handball team. This study was performed following the ethical standards of the Helsinki Declaration and the participants signed an informed consent form. All the participants underwent two balance assessments. One assessment occurred prior to their COVID-19 infection and the second assessment was administered following the infection. The second (i.e., post-COVID) assessment was conducted approximately one month following the completion of their COVID quarantine. It is worth noting that all the athletes on the professional handball team had their balance assessed in 2019 per their involvement in a separate research study. As the COVID-19 pandemic spread through the handball team in 2020 and 2021, athletes who were infected with the virus were then recruited to participate in the present study.

All the participants performed an individual 3 min warm-up. The Romberg test was used to investigate the effect COVID-19 had on static balance [28,29,30]. This test is an appropriate tool to diagnose sensory ataxia: a gait disturbance caused by abnormal proprioception involving information about the locations of joints [31]. It has been used in clinics for 150 years; its ability to objectively test the relationship between human bipedal locomotion and the vestibular system has been verified several times [31,32]. It is also proven to be a sensitive and accurate means of measuring the degree of disequilibrium caused by central vertigo, peripheral vertigo, and head trauma [33,34]. The Romberg test’s ability to gauge true proprioception status can be confounded by the vestibular and vision somatosensory system, which may compensate with vestibular function and vision. The Romberg test sign removes the visual and vestibular components that contribute to maintaining balance; it can thus specifically identify a proprioception-related neurologic disease. The Romberg test consists of standing with the individual’s feet together and upper extremities extended; so the arms are parallel to the floor. The participants completed this test for 10 s with their eyes opened and 10 s with their eyes closed. The tests were performed in the same sports uniform the athletes wore during match play.

The measurements were made using a Kistler force plate (Type 2812A1-3). Ground reaction forces were collected with 100 Hz and band-pass filtered. Data from the force plate were processed and calculated by BioWare^®^ Software 4.0 (Kistler Holding AG., Winterthur, Switzerland). Data from the force platform were processed to obtain postural parameters about the center of pressure (COP) displacements. The following parameters in the anteroposterior (AP) and mediolateral (ML) axes were analyzed: maximum excursion of COP along the axes (RAP and RML); mean velocity of COP displacements along the axes (MVAP and MVML); and work distribution along the axes (WAP and WML).

### 2.3. Statistical Analysis

The normality of distribution was checked with the Shapiro–Wilk test (StatSoft, Inc. STATISTICA version 13.0) and a normal distribution was found. To fully analyze our data, we conducted a 2 (condition: COVID, no-COVID) X 2 (test: pre-test, post-test) ANOVA with repeated measures on the second factor after Bonferroni adjustments. All the differences between conditions were calculated using the Fisher’s posthoc NIR test. All the statistical analyses were performed using STATISTICA software (TIBCO Software Inc., 2017, Palo Alto, CA, USA). Statistica (data analysis software system), version 13. http://statistica.io (accessed on 29 March 2022).

## 3. Results

ANOVA proved the COVID-19 effect for RAP (F(1,10) = 69.910, *p* < 0.001, η2 = 0.875), RML (F(1,10) = 76.359, *p* < 0.001, η2 = 0.884), MVAP (F(1,10) = 23.209, *p* < 0.001, η2 = 0.699), MVML (F(1,10) = 26.670, *p* < 0.001, η2 = 0.735), as well as WAP (F(1,10) = 216.741, *p* < 0.001, η2 = 0.956) and WML (F(1,10) = 375.479, *p* < 0.001, η2 = 0.974) (Figure 1, Figure 2 and Figure 3). 

The Fisher’s posthoc NIR test showed that the results in groups E and C did not differ before the COVID-19 pandemic. After the end of the disease, all the parameters describing COM displacement worsened significantly in the athletes in group E (*p* < 0.001). The values in group E were also significantly worse than the results in group C (RAP—*p* < 0.001, RML—*p* < 0.001, MVAP—*p* < 0.01, MVML—*p* < 0.01, WAP—*p* < 0.01 and WML—*p* < 0.05) (Figure 1, Figure 2 and Figure 3).

## 4. Discussion

COVID-19 is highly transmittable in sporting environments due to its viability, long incubation period, and milder symptoms, especially in contact sports. The final impact of the COVID-19 pandemic on sports and exercise cannot be determined at this point; however, the information that we reported here may provide valuable guidance to athletes and governing committees to move forward safely. There is a distinct need to develop and adopt consistent measures for the resuming of sports activities, including training and competition, in a way that places the health and well-being of athletes at the forefront while protecting coaches, allied staff, and spectators.

The purpose of this study was to evaluate the effect of COVID-19 on maintaining balance in skilled handball players. The impact of SARS-CoV-2 infections on the central and peripheral nervous systems has recently been questioned. Similarly, the effects on the audio-vestibular system have been examined as well [27]. These effects likely contribute to the symptoms commonly observed during or following a COVID-19 infection. For example, balance problems are one of the common symptoms that have been self-reported by infected persons, even though there are currently no objective data to confirm this. In our study, we found that during the performance of the Romberg test, the COP parameters were significantly worse (*p* < 0.05) following a COVID infection. These results are consistent with the self-reported evidence provided by patients. In Yılmaz et. al. [12], approximately 30% of the patients complained of balance problems; these were mainly in the form of dizziness [12]. Our results provide the first empirical evidence to support these claims.

Moreover, coronavirus infections are accompanied by common and uncommon respiratory symptoms. Celik and colleagues [7] measured the percent of maximal inspiratory and expiratory pressure values within athletes who tested positive for COVID-19; they found that these measures were statistically lower than in athletes without a positive COVID-19 infection (*p* <  0.05). Dynamic lung volumes were similar in groups (*p*  >  0.05). Inspiratory and expiratory muscle strength in athletes following COVID-19 were more affected compared to athletes without a COVID-19 infection. Pulmonary functions were mostly preserved in athletes with and without confirmed infection. Given that respiratory muscles contribute to the individual’s ability to maintain balance [35], the decrease in balance found within this study could be due to a potential weakening of respiratory muscles previously found following a COVID-19 infection [7]. This hypothesis should be investigated in future research.

Another explanation for this observed decrease in balance performance could be indirectly due to the audio-vestibular effects of a cellular infection within the inner ear as a result of the COVID-19 virus [8]. The specific cells that COVID-19 has been found to infect and kill within the ear have been proposed to explain the negative hearing and loss of balance side effects commonly reported [8] by COVID-19 patients. Additionally, this decrease in balance could be a consequence of the neurological symptoms experienced during COVID-19; these include dizziness and loss of balance [4,24]. These neurological symptoms, which have also been found within previous coronaviruses [25], have been proposed to be due to a compromised central nervous system during or following COVID-19 [24,26]. Therefore, the introduction of balance exercises to improve proprioception and to train the brain so that it can recognize the position of a body segment at all times seems reasonable. Thus, a balance exercise program will train and facilitate proprioception pathways under competitive circumstances effectively. Specifically, to prevent limb injuries, peripheral and central nervous system receptors, mechanoreceptors within muscles, tendons, and ligaments have to be activated and guide the body segments to move properly [17]. Thus, the goal of balance exercises possibly reduces the time between the neural stimuli and muscular response preventing injuries [12].

Demonstrating a relationship between COVID-19 and deterioration in balance requires further scientific research. However, coaches, teachers, athletic trainers, medical doctors, and physiotherapists should be particularly aware of the possibility of such a relationship. The literature discussing the possibility of a relationship between dizziness, balance, and COVID-19 is very poor and almost completely devoid of objective data describing this loss of motor function. An additional complication is that under the term “vertigo” there are many clinical forms, including the subjective sensation of movement (spinning), sensation close to fainting, feeling of static uncertainty, and instability of the ground. Due to the complex structure of the balance system, such as the vagus, vestibular nerve, vestibular nuclei, and the complex centers and connections between them in the cerebellum, brainstem, and cortex, the organ of vision, deep sensory receptors in the tendons, muscles, ligaments, and joints, as well as the complicated mechanisms of its operation, linking the occurrence of symptoms of its damage (dizziness) with the influence of any external factor, including COVID-19, will be very difficult.

## 5. Strengths and Limitations

With the recent COVID-19 pandemic, there is a lot of research into the effects of the virus on sports performance. To the best of our knowledge, however, no existing studies are looking at the direct equilibrium effects of COVID-19 contamination. All the results were evaluated using reliable and valid measurement methods under standardized conditions. The results of this study are conclusive; thus, our findings could provide important information for managing sports clubs in the context of countering the future effects of the COVID-19 pandemic. Moreover, these results could form the basis of an extensive prospective study of the effects of COVID-19 infection on physical performance. However, we are aware that apart from the COVID-19 infection factor, it is also possible that the players reacted differently to the temporary lack of training. Moreover, although the time of exclusion from training was the same, the reactions could be individually varied; and due to the small size of the group, we were not able to observe it. Therefore, in the future, this research should be carried out on a larger group; it should also take into account the detraining factor, which has not been assessed here.

## 6. Conclusions

In conclusion, the results we reported here are the first to show an objectively measured association between COVID-19 and the ability to maintain static balance. Further intensive multifaceted research on this issue is needed. Future research should continue this investigation and compare balance in individuals after a COVID-19 infection, and those who have not had COVID-19. Additionally, it would be valuable to understand whether these balance issues are temporarily observed or long-lasting. These results show that following self-isolation after a COVID-19 infection, there were decreases in balance performance. It will be important to understand how long these observations last and compare them to others without infection.

## Figures and Tables

**Figure 1 ijerph-19-10470-f001:**
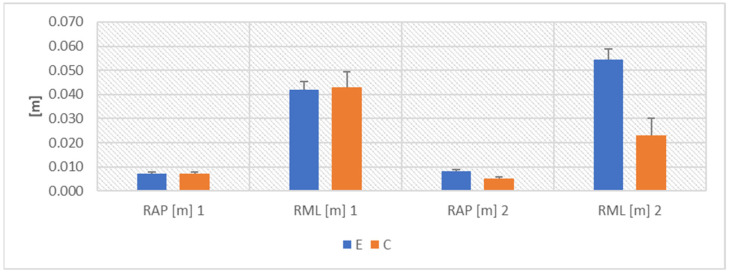
Mean values ± SD of COM displacement.

**Figure 2 ijerph-19-10470-f002:**
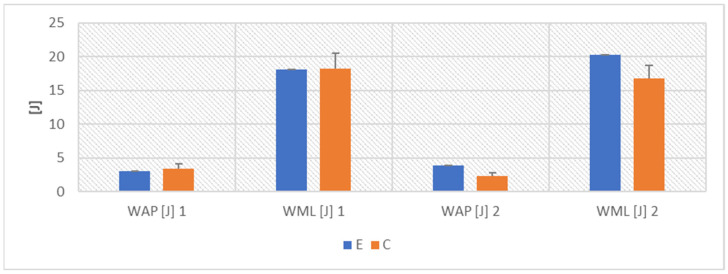
Mean values ± SD of COM work.

**Figure 3 ijerph-19-10470-f003:**
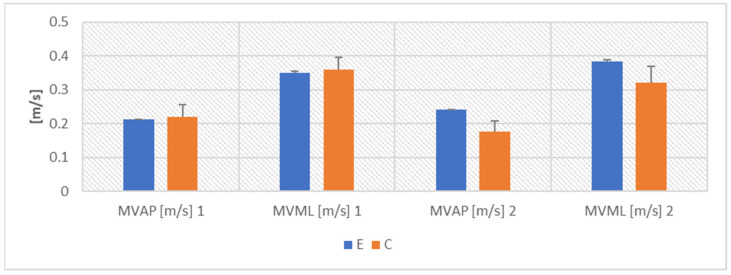
Mean values ± SD of COM velocity.

## Data Availability

The data presented in this study are available on request from the corresponding author.

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
