# Peer review of "Effect of Coronavirus 19 on Maintaining Balance in Skilled Athletes"

_ijerph, 2022, doi:10.3390/ijerph191710470_

Round 1

Reviewer 1 Report

GENERAL COMMENT:

Thank you for the opportunity to review this interesting issue and we value the effort that you put into this study. I think that this research is able to get the reader interested and work together on the topic. I think this article has some potential but there are some critical flaws as described below.

-Is static balance important in a handball player? I think dynamic balance will be more important. What do you think?

SPECIFIC COMMENTS

1. Abstract

-Is static balance important in a handball player? I think dynamic balance will be more important. If possible, add a dynamic balance.

-Rewrite keywords

2. Introduction

-Appropriate, but it would be better if you mention how COVID-19 affects athletes. Also, it would be good for understanding the purpose of this study to mention the important of static balance in athletes.   3. Material

-Same as the previous question. Is static balance important in a handball player? In particular, is it possible to evaluate the balance ability of a handball player with the romberg test? Also, is the balance test time of 10 seconds appropriate in athletes? In addition, random statements without backed by any citation or evidence.

-Be more specific about the test method and measurement parameters.

-The test in this study was conducted at one-month after the COVID-19 infection. What changes do you think will happen after that? On Line 51 of this study, the long-term effects of COVID-19 are currently unclear.

- There are variety factors that affect balance, such as ankle sprains, cruciate ligament injuries, and low back pain. How were these considered?

- Is the conclusion in the last paragraph of the discussion, but why is the conclusion section separate?

4. Conclusion

-Respiration-related systems were not evaluated in this study. Therefore, in the conclusion section, it is not appropriate to explain the conclusion of this study. Rewrite the conclusions

-What is the clinical relevance of this study? need to breathing exercises? need to balance exercises?

5. References

-It doesn't fit the form. Rewrite the references

Author Response

RESPONSE TO REVIEWERS' COMMENTS

Dear Editor and Reviewers,

The authors would like to thank the reviewers for their precious time and invaluable comments. We have carefully addressed all the comments. The corresponding changes and refinements made in the revised paper are summarized in our response below. We hope that the modifications and explanations will be acceptable to you.

REVIEWER #1

Dear Authors

Thank you for the opportunity to review this interesting issue and we value the effort that you put into this study. I think that this research is able to get the reader interested and work together on the topic. I think this article has some potential but there are some critical flaws as described below.

Is static balance important in a handball player? I think dynamic balance will be more important. If possible, add a dynamic balance.

Response 1: Thank you for this suggestion. We have decided to analyze only static balance in handball players. The jumping, landing, and pivoting activities contained in team ball have been described as injury risk factors by many authors proposing prevention strategies based on improving athletes' balance ability (Hewett 2001; Junge 2002). Also, the rehabilitation bibliography supports that balance exercise programs may improve proprioceptive function not only during rehabilitation but also during the competition period, protecting athletes from forthcoming injuries effectively (Hoffman & Payne, 1995; Wedderkopp et al., 1999; McHugh et al., 2007).

According to reviewers’ suggestions Rewrite keywords

Response 2: We agree with the reviewer and have made corrections.

Appropriate, but it would be better if you mention how COVID-19 affects athletes. Also, it would be good for understanding the purpose of this study to mention the important of static balance in athletes.   

Response 3: We have made modifications and now the paper does offer additional information based on previous work which has been published on this topic.

Same as the previous question. Is static balance important in a handball player? In particular, is it possible to evaluate the balance ability of a handball player with the Romberg test? Also, is the balance test time of 10 seconds appropriate in athletes? In addition, random statements without backed by any citation or evidence.

Response 4: The Romberg test is an appropriate tool to diagnose sensory ataxia, a gait disturbance caused by abnormal proprioception involving information about the locations of joints [1]. It has been used in clinics for 150 years, however, and its ability to objectively test the relationship between human bipedal locomotion and the vestibular system has been verified several times [1, 2]. It is also proven to be a sensitive and accurate means of measuring the degree of disequilibrium caused by central vertigo, peripheral vertigo, and head trauma [3, 4].

  1. El-Kashlan, H.K., Shepard, N.T., Asher, A.M., Smith-Wheelock, M., Telian, S.A.: Evaluation of clinical measures of equilibrium. Laryngoscope 108, 311–3119 (1998)
  2. Ford-Smith, C.D., Wyman, J.F., Elswick, R.K.J., Fernandez, T., Newton, R.A.: Test-retest reliability of the sensory organization test in noninstitutionalized older adults. Arch. Phys. Med. Rehabil. 76, 77–81 (1995)
  3. Schippati, M., Nardone, A.: Free and supposed stance in Parkinson’s disease: The effect of posture and ‘postural set’ on leg muscle responses to perturbation, and its relation to the severity of the disease. Brain 114, 1227–1244 (1991)
  4. Black, F.O., Peturka, J.H., Shupert, C.L., Nashner, L.M.: Effects of unilateral loss on vestibular function on vestibulo-ocular reflex and postural control. Otol. Rhinol. Laryngol 98, 884–889 (1989)

Be more specific about the test method and measurement parameters.

Response 5: We agree with the reviewer and we have added the information about the test method and measurement parameters in the methodology.

The test in this study was conducted at one-month after the COVID-19 infection. What changes do you think will happen after that? On Line 51 of this study, the long-term effects of COVID-19 are currently unclear.

Response 6: We agree with the reviewer – the long-term effects of COVID-19 are currently unclear. We predicted the infected players after illness will have a problem with maintaining their balance.

There are variety factors that affect balance, such as ankle sprains, cruciate ligament injuries, and low back pain. How were these considered?

Response 7: We agree with the reviewer and we have added the information about the appropriate inclusion and exclusion criteria in the methodology when designing a study.

Is the conclusion in the last paragraph of the discussion, but why is the conclusion section separate?

Response 8: We agree with the reviewer and we have changed this part of the discussion. The conclusion section is separate because these are the requirements of the journal.

Respiration-related systems were not evaluated in this study. Therefore, in the conclusion section, it is not appropriate to explain the conclusion of this study. Rewrite the conclusions.

Response 9: We agree with the reviewer and we have changed this part of the discussion.

What is the clinical relevance of this study? need to breathing exercises? need to balance exercises?

Response 10: We think there is a sure signal for coaches to use and apply adequate exercises depending on the case of the disease and its complications in the players.

It doesn't fit the form. Rewrite the references.

Response 10: We agree with the reviewer and have made corrections.

REVIEWER #2

The authors investigated the effect of COVID-19 on maintaining static balance amongst highly skilled handball players. It was postulated from the study findings that COVID-19 may have an impact on maintaining the static balance of the players. Overall, it is an interesting work and I congratulate the authors for their effort. However, many issues hinder the publication of the manuscript in its current form.

More appropriate keywords should be provided.

Response 1: We agree with the reviewer and have made corrections.

L121. Which replace with Who

Response 2: This has been changed.

The authors highlighted that the data gathered was not normally distributed (L138-139). I am wondering why then the authors applied the ANOVA test to the data which is a parametric test. I think considering the sample size of the study (relatively small) coupled with the non-normality of the data, the authors ought to have applied a non-parametric test e.g., the Friedman test

Response 3: Thank you for this suggestion. The data gathered was normally distributed and therefore applied the ANOVA test to the data which is a parametric test.

.     What E and C refer to in Figures 1-3? If it refers to the grouping of the participants, then the explanation of the grouping should be first introduced before the Figures. The naming (E and C) is also not consistent. Figure 2 indicated C, C.

Response 4: We agree with the reviewer and have made corrections.

The decimal points for the p-values should be consistent

Response 5: This has been changed.

L200-202- It is premature to make such claims since your participants did not complain of any dizziness or hearing loss.

Response 6: We agree with the reviewer to make such claims is premature, but we are only to make such predictions based on other research.

The study is subject to many limitations that the authors did not consider/report. Firstly, the reduced balance observed from the previously infected players might be a result of stoppage/reduction of training during the infection period/ quarantine. In other words, the observed changes may be fitness related. Secondly, the authors did not report the acquisition of other important and related historical data of the infected players. For instance, complaint-related data comprising of any issue of hearing loss, dizziness etc. were not collected or reported herein. Thus, it is rather misleading to assume that these variables could have an impact on the players as speculated in the discussion.

Response 7: With the recent COVID-19 pandemic, there is a lot of research into the effects of the virus on sports performance. To the best of our knowledge, however, no existing studies are looking at the direct equilibrium effects of COVID-19 contamination. All results were evaluated using reliable and valid measurement methods under standardized conditions. The results of this study are conclusive, so our findings could provide important information for managing sports clubs in the context of countering the future effects of the COVID-19 pandemic. Moreover, these results could form the basis of an extensive prospective study of the effects of COVID-19 infection on physical performance. However, we are aware that apart from the COVID-19 infection factor, it is also possible that the players reacted differently to the temporary lack of training. And although the time of exclusion from training was the same, the reactions can be individually varied and due to the small size of the group, we were not able to observe it. Therefore, this research should be carried out in the future on a larger group, also taking into account the detraining factor which has not been assessed here.

Reviewer 2 Report

The authors investigated the effect of COVID-19 on maintaining static balance amongst highly skilled handball players. It was postulated from the study findings that COVID-19 may have an impact on maintaining the static balance of the players. Overall, it is an interesting work and I congratulate the authors for their effort. However, many issues hinder the publication of the manuscript in its current form.

1.       Keywords

More appropriate keywords should be provided

Introduction

2.        L121. Which replace with Who

Statistical Analysis

3.       The authors highlighted that the data gathered was not normally distributed (L138-139). I am wondering why then the authors applied the ANOVA test to the data which is a parametric test. I think considering the sample size of the study (relatively small) coupled with the non-normality of the data, the authors ought to have applied a non-parametric test e.g., the Friedman test

Results.

4.       What E and C refer to in Figures 1-3? If it refers to the grouping of the participants, then the explanation of the grouping should be first introduced before the Figures. The naming (E and C) is also not consistent. Figure 2 indicated C, C.

5.       The decimal points for the p-values should be consistent

Discussion

6.       L200-202- It is premature to make such claims since your participants did not complain of any dizziness or hearing loss.

Main concern

7.       The study is subject to many limitations that the authors did not consider/report. Firstly, the reduced balance observed from the previously infected players might be a result of stoppage/reduction of training during the infection period/ quarantine. In other words, the observed changes may be fitness related. Secondly, the authors did not report the acquisition of other important and related historical data of the infected players. For instance, complaint-related data comprising of any issue of hearing loss, dizziness etc. were not collected or reported herein. Thus, it is rather misleading to assume that these variables could have an impact on the players as speculated in the discussion.

Author Response

(The authors gave the same response as above.)

Round 2

Reviewer 1 Report

1. Response 1: Thank you for this suggestion. We have decided to analyze only static balance in handball players. The jumping, landing, and pivoting activities contained in team ball have been described as injury risk factors by many authors proposing prevention strategies based on improving athletes' balance ability (Hewett 2001; Junge 2002). Also, the rehabilitation bibliography supports that balance exercise programs may improve proprioceptive function not only during rehabilitation but also during the competition period, protecting athletes from forthcoming injuries effectively (Hoffman & Payne, 1995; Wedderkopp et al., 1999; McHugh et al., 2007).

Reply: I can't find anywhere in your suggestion that static balance is important in athletes. Please provide a reference for this.

2. Response 4: The Romberg test is an appropriate tool to diagnose sensory ataxia, a gait disturbance caused by abnormal proprioception involving information about the locations of joints [1]. It has been used in clinics for 150 years, however, and its ability to objectively test the relationship between human bipedal locomotion and the vestibular system has been verified several times [1, 2]. It is also proven to be a sensitive and accurate means of measuring the degree of disequilibrium caused by central vertigo, peripheral vertigo, and head trauma [3, 4].

  1. El-Kashlan, H.K., Shepard, N.T., Asher, A.M., Smith-Wheelock, M., Telian, S.A.: Evaluation of clinical measures of equilibrium. Laryngoscope 108, 311–3119 (1998)

  2. Ford-Smith, C.D., Wyman, J.F., Elswick, R.K.J., Fernandez, T., Newton, R.A.: Test-retest reliability of the sensory organization test in noninstitutionalized older adults. Arch. Phys. Med. Rehabil. 76, 77–81 (1995)

  3. Schippati, M., Nardone, A.: Free and supposed stance in Parkinson’s disease: The effect of posture and ‘postural set’ on leg muscle responses to perturbation, and its relation to the severity of the disease. Brain 114, 1227–1244 (1991)

  4. Black, F.O., Peturka, J.H., Shupert, C.L., Nashner, L.M.: Effects of unilateral loss on vestibular function on vestibulo-ocular reflex and postural control. Otol. Rhinol. Laryngol 98, 884–889 (1989)

Reply: Nowhere did your reference to the Romberg test for assessing static balance in athletes.

Only parkinson’s and older adults in your references

Please provide a reference to the Romberg test in athletes.

Author Response

REVIEWER #1

Dear Authors

Thank you for the opportunity to review this interesting issue and we value the effort that you put into this study. I think that this research is able to get the reader interested and work together on the topic. I think this article has some potential but there are some critical flaws as described below.

I can't find anywhere in your suggestion that static balance is important in athletes. Please provide a reference for this.

Response 1: We agree with the reviewer and we have added the information that static balance is important in athletes. The reference has been added.

Nowhere did your reference to the Romberg test for assessing static balance in athletes.

Response 2: We agree with the reviewer and we have made corrections.

Reviewer 2 Report

The manuscript has greatly improved from the initial iteration. However, in the limitation section of the manuscript, the authors should highlight those relevant variables that they are unable to capture in their investigations(Historical data on complaints related issues after COVID-19 infection). This could guide provide a guide for future studies.

Author Response

REVIEWER #2

The manuscript has greatly improved from the initial iteration. However, in the limitation section of the manuscript, the authors should highlight those relevant variables that they are unable to capture in their investigations (Historical data on complaints related issues after COVID-19 infection). This could guide provide a guide for future studies.

Response 1: We agree with the reviewer and have made corrections.

Moderate English changes required

Response 2: According to reviewers’ suggestions, manuscript has been reviewed and edited by a university native English speaker.
